# Exploring Aromatic *S*-Thioformates as Photoinitiators

**DOI:** 10.3390/polym15071647

**Published:** 2023-03-26

**Authors:** Paul Rieger, Sabrina Pueschmann, Michael Haas, Max Schmallegger, Gema Guedes de la Cruz, Thomas Griesser

**Affiliations:** 1Institute of Chemistry of Polymeric Materials, Montanuniversität Leoben, Otto Glöckelstrasse 2, 8700 Leoben, Austria; 2Institute of Inorganic Chemistry, Graz University of Technology, Stremayrgasse 9, 8010 Graz, Austria; 3Institute of Physical and Theoretical Chemistry, Graz University of Technology, Stremayrgasse 9, 8010 Graz, Austria

**Keywords:** photochemistry, photoinitiator, thiol-ene

## Abstract

Thiyl radicals were generated from aromatic *S*-thioformates by photolysis. The corresponding photo-initiated decarbonylation allows initiating polymerization reactions in both acrylate- and thiol-acrylate-based resin systems. Compared to aromatic thiols, the introduction of the photolabile formyl group prevents undesired reactions with acrylate monomers allowing photoinitiators (PIs) with constant reactivity over storage. To demonstrate the potential of *S*-thioformates as PIs, the bifunctional molecule *S,S′*-(thiobis(4,1-phenylene))dimethanethioate (**2b**) was synthesized, providing reactivity under visible light excitation. Consequently, acrylate-based formulations could successfully be processed by digital light processing (DLP)-based stereolithography at 405 nm in high resolution.

## 1. Introduction

The photochemistry of aromatic *O*- and *N*-acyl derivatives has been intensively studied in polymeric materials over the last 3 decades [1,2,3,4,5]. The conversion of arylesters to *ortho*- and *para*-hydroxyketones by the photo-Fries rearrangement was first observed by Anderson and Reese in the 1960s [6]. Later, similar photoreactions were also described for anilides, aryl carbonates, carbamates, sulfonates, sulfamates, thioesters, sulfonanilides, and sulfenanilides [7,8].

Since the discovery, it has taken a long time to elucidate the mechanism of the photo-Fries rearrangement. The generally accepted mechanism of the photoreaction of aryl esters proceeds via free radical intermediates, whereby a cleavage of the C-O bond from the excited singlet state (S1) occurs in a first step [9]. The radicals generated by the photolysis can recombine and then yield *ortho-* and *para*-hydroxyketones “cage product”. 

Due to the formation of stable photoproducts with high absorption in the UV range (internal filter effect) the conversion of aryl ester groups in polymeric materials is limited to 20–30% [2,10,11,12]. With the aim of achieving higher conversion rates and yields, the photoreactions of aromatic esters of formic acid and *N*-formamides has been investigated [13]. Frechet and Tessier have shown that irradiation of poly[*p*-(formyloxy)styrene] leads to the formation of poly(*p*-hydroxystyrene) in high yield. In this case, the formyl radical formed decomposes rapidly in the solvent cage to carbon monoxide, preventing recombination of the radical pair.

Interestingly, there are only a few reports that deal with the photoreactivity of aromatic *S*-acyl derivatives [8]. In this reaction, the cleavage of the C-S bond leads to the formation of aromatic thiyl radicals, which mainly recombine in the “solvent cage” to the corresponding disulfides, while the acyl radicals decarbonylate.

In general, the chemistry of light-generated thiyl radicals has been intensively studied over the last 2 decades in the context of the radical-mediated thiol–ene and thiol–yne coupling reaction [14,15,16,17,18,19,20,21,22,23,24,25].

Very recently, the group of C. Bowman showed that the unique photodynamics of aromatic mercaptans allow their employment as effective visible light photoinitiators by the photolysis of the S-H bond [26]. They demonstrated that aromatic thiols are useful initiators in an acrylate polymerization, bulk thiol-ene polymerizations, solventless and initiatorless small molecule thiol–ene reactions, and thiol–ene hydrogel polymerization reactions with irradiation wavelengths up to 405 nm. Since these initiators will incorporate into polymer networks, preventing leaching of unreacted PI, aromatic thiols are highly interesting for the realization of biocompatible photopolymers. In addition, the mercapto groups of the PI can reduce oxygen inhibition in (meth)acylate polymerisation, which decreases the amount of non-polymerized monomers in the cured materials.

However, it is well reported that thiols can react with ene monomers also in the dark, limiting the shelf-life stability of thiol–ene resins significantly [27,28]. Several initiation mechanisms are discussed in the scientific literature, including (1) the decomposition of peroxide impurities and subsequent initiation of a thermal free-radical reaction, (2) the reaction of hydroperoxide impurities to form thiyl radical intermediates, (3) thiol–ene reactions due to a base-catalyzed nucleophilic addition, or (4) the spontaneous initiation of polymerization via the generation of radicals through a ground-state charge transfer complex formed between the thiol and the ene components. In the context of thiol-based PIs, a limited stability of mercapto groups in (meth)acrylate-based resins can reduce their initiation efficiency with storage time.

Herein, the photochemistry of *S*-thioformates is investigated with the aim to exploit the UV-induced photodecarbonylation reaction to generate thiyl radicals capable of initiating polymerization reactions in acrylate- as well as thiol-acrylate-based resin systems. The introduction of the photolabile formyl group prevents dark reactions, promising mercapto-based PIs with constant reactivity over storage time. 

## 2. Results and Discussion

### 2.1. Photoreactivity of S-Phenyl Thioformate

*S*-phenyl thioformate was synthesized using a mixture of formic acid and acetic anhydride (yield: 90%) as described elsewhere [29]. The photoreaction of *S*-phenyl thioformate was investigated by ^1^H-NMR spectroscopy (see Appendix A) in CD_3_CN (48.2 mM). Figure 1 shows the conversion of *S*-phenyl thioformate during illumination with monochromatic UV light (254 nm, 5.08 mW cm^−2^). In these studies, the decrease in the singlet H signal of the thioformate group (**H**COS-, 10.24 ppm) was followed revealing a conversion of 94% after 60 min of irradiation. The formed thiyl radicals can react to a complex product mixture as reported by Grunwell et al. [8] The investigation of the photoproducts was carried out by HPLC, determining diphenyl disulfide (appox. 40%) as the main product. The area of the UV-Vis detector signals was used to estimate the composition of the product mixture. Besides the educt, i.e., *S*-phenyl thioformate (approx. 10%), and benzenethiol (approx. 10%), several other photoproducts, which have not been identified, were detected (see Appendix A). 

The mechanism of the photoreaction was established by spin-trapping electron paramagnetic resonance (ST-EPR) spectroscopy, using 5,5-dimethyl-1-pyrroline-N-oxide (DMPO) as the trapping reagent. Figure 2 (left, grey spectrum) shows the EPR spectrum obtained upon irradiation of S-phenyl thioformate in the presence of DMPO. The hyperfine data used to simulate the experimental EPR spectra (Figure 2) reveal the formation of thiyl (PhS·), formyl (CHO·), and hydrogen radicals (H·, for details, see Appendix A) [30,31,32].

In a further experiment, the formation of carbon monoxide was proven by analyzing the formed gas by GC–MS coupling (Appendix A), confirming the proposed photodecarbonylation mechanism shown in Figure 2 (right). The hydrogen radical formed is assumed to undergo recombination reactions with the thiyl radical and the formyl radical, resulting in the formation of benzenethiol and formaldehyde, respectively.

### 2.2. Photoinitiation of Acrylate Polymerization

Following the idea to exploit the photogenerated thiyl radicals for the initiation of monomer polymerization, *S*-phenyl thioformate was added in a concentration of 1 mM to hexyl acrylate as model monomer. For comparison, the initiation behavior of thiophenol (1 mM) as well as diphenyl disulfide (1 mM) was additionally studied in hexyl acrylate. Both of those molecules are known to form thiyl radicals either by S-H or S-S fission upon UV irradiation and have already been used for the initiation of acrylate polymerization [26].

The S-C, S-S, and S-H photocleavage reactions were initiated using multichromatic UV-light (185–665 nm, 58.7 mW cm^−2^). While *S*-phenyl thioformate and diphenyl disulfide show a similar initiation performance, leading to an acrylate conversion of 98 and 95%, respectively, only 89% were reached with thiophenol, as depicted in Figure 3 (left). In this context, it is important to mention that thiophenol provides the highest UV absorption (in the range 290–360 nm, as shown in Appendix A) among these molecules. From these data, it can be concluded that the effectiveness of photolysis follows S-C > S-S > S-H. 

With the aim to initiate photopolymerization under visible light, the bifunctional PI *S,S′*-(thio*bis*(4,1-phenylene))dimethanethioate (**2b**, see Figure 4, left) was synthesized as described in the Appendix A (yield: 91%). This compound was chosen as the corresponding thiol 4,4′- thio*bis*benzenethiol (**2a**) provides reasonable initiation performance under illumination of 405 nm [26]. Similar to *S*-phenyl thioformate, an illumination of **2b** with visible light of 405 nm leads to a decarbonylation reaction (conversion of 42%), as shown in Figure 4 (left) and Appendix A). To evaluate and compare the efficiency of thiyl radical formation, **2a** and **2b** were added in a concentration of 1 mM to trimethylolpropane trimethacrylate (TMPTMA) as a model monomer. Although **2b** shows a higher initial polymerization rate, similar double bond (DB) conversions (25% vs. 28%) were obtained upon illumination with light of 405 nm, as shown in Figure 4 (middle). As in the case of *S*-phenyl thioformate, the formylated compound **2b** shows a lower UV-Vis absorption, as revealed in Appendix A. The generally lower DB conversions are explained by the chosen model monomer, which reaches the glassy state faster than the monofunctional hexyl acrylate.

An important prerequisite for photoinitiators is their stability in the resin formulation. Consequently, the change in the initiation performance of **2a** and **2b**, respectively, in TMPTMA during storage at 50 °C in the absence of light was investigated. As shown in Figure 4 (middle), the reactivity of **2a** (black lines) decreases significantly after 7 days of storage (DB conversion: 8%), while the performance of **2b** (red lines) is hardly affected upon illumination with visible light of 405 nm (DB conversion: 24%). This behavior is explained by the consumption of the mercapto groups of the aromatic thiol **2a** by dark reactions. It is well reported that basic as well as peroxide and hydroperoxide impurities can lead to the formation of thiolate and thiyl radicals, respectively, which react with the ene monomer to the corresponding thioether and thus are mainly responsible for the instability of thiol–ene resins [33]. In order to supress radical initiated dark reaction, propyl gallate has been added to the resin formulation. It is known that radical scavengers can reduce dark reactions, but not completely prevent them [33].

In particular, due to the higher pk_a_ of aromatic mercapto compounds (pk_a_ (thiophenol): 6.62 vs. pk_a_ (propanthiol): 10.6) and the resonance stabilization of aromatic thiyl radicals, it can be expected that these compounds are more prone to undergo dark reactions compared to aliphatic thiols. 

Importantly, the formyl protective group of **2b** offers an almost constant reactivity over storage time, as shown in Figure 4 (middle, red curves). Additionally, a formulation without PI (green line) shows no reactivity under illumination with visible light of 405 nm that excludes direct excitation of the acrylate monomer at this wavelength.

To demonstrate the potential of *S*-thioformates as photoinitiators, 3D printing experiments were carried out using DLP-based stereolithography. In these investigations, 0.01 wt% Sudan II G were added as light absorber to the formulation of TMPTMA containing 0.5 wt% pyrogallol as stabilizer. Although a dose of 33.43 mJ cm^−2^ was required for the curing of a single layer, the investigated formulation could be printed successfully in high resolution, as shown on the test structures illustrated in Figure 4 (right). The relatively low sensitivity of the resin can be attributed to the lower initiation performance of aromatic *S*-thioformates and thiols compared to type I PIs such as hydroxyketones or phosphine oxides.

### 2.3. Photoinitiation of Thiol-Methacrylate Polymerization

To evaluate and compare the initiation efficiency of aromatic *S*-thioformates and thiols in a thiol–ene reactive system, **2a** and **2b** were added in a concentration of 1 mM to a TMPTMA- pentaerythrit-tetrakis-(3-mercapto-propionat) (PETMP) formulation ([ene]:[SH] = 70 mol%:30 mol%) containing 0.5 wt% propyl gallate as radical scavenger. As shown in Figure 5 left, both PIs provide similar initiation performance in the chosen system. After illumination with a dose of 57.1 mJ cm^−2^, a DB conversion of 68% was obtained in both cases. 

It is important to note that the addition of the aromatic thiol to the TMPTMA/PETMP significantly reduces the stability of the formulation, as shown by the rheological measurements in Figure 5 (right). While the resin containing **2b** shows no significant change in storage modulus during the period considered (125 min) at 40 °C, the addition of **2a** destabilizes the thiol–ene system inducing gelation after 67 min.

This behavior can again be explained by the comparably high pk_a_ of aromatic mercapto compounds and the resonance stabilization of aromatic thiyl radicals. The added aromatic thiols are more susceptible to the formation of thiyl radicals and thiolate anions, which can initiate thiol–ene and thiol–Michael polymerisation reactions, respectively.

## 3. Conclusions

In conclusion, the photoreaction of aromatic *S*-thioformates was investigated, which showed a decarbonylation reactions upon excitation with UV light. Based on the idea of using light-generated thiyl radicals to initiate monomer polymerization, the initiation performance of *S*-phenyl thioformate was investigated in comparison with thiophenol and diphenyl disulfide. These experiments revealed the superiority of the C-S fission in hexyl acrylate. With the aim of initiating photopolymerization under visible light, the bifunctional PI **2b** was synthesized, providing similar initiation performance as **2a** in an TMPTMA- and PETMP-TMPTMA-based formulation, respectively. To highlight the potential of **2b**, acrylic formulations were processed by DLP-based stereolithography using light of 405 nm. Although a comparatively high dose of 33.43 mJ cm^−2^ was required to cure a single layer, TMPTMA was successfully printed at high resolution. 

Due to the high pk_a_ of aromatic thiols and the resonance stabilization of aromatic thiyl radicals, these compounds are prone to undergo dark reactions with acrylic monomers. The introduction of the photolabile formyl group prevents undesired reactions with acrylate monomers allowing mercapto based PIs with constant reactivity over storage.

## Figures and Tables

**Figure 1 polymers-15-01647-f001:**
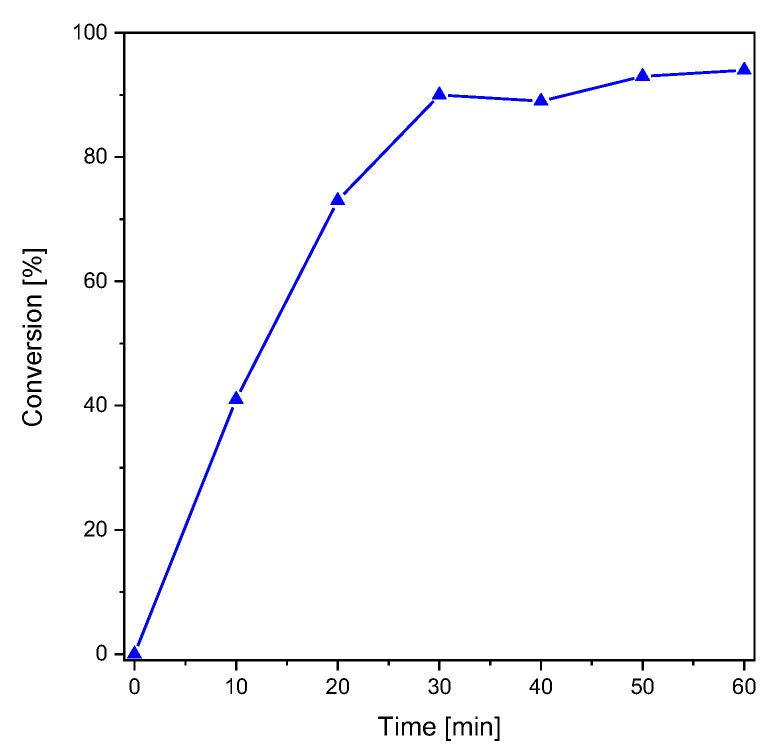
Conversion of S-phenyl thioformate under UV illumination (right, 254 nm, 5.08 mW cm^−2^).

**Figure 2 polymers-15-01647-f002:**
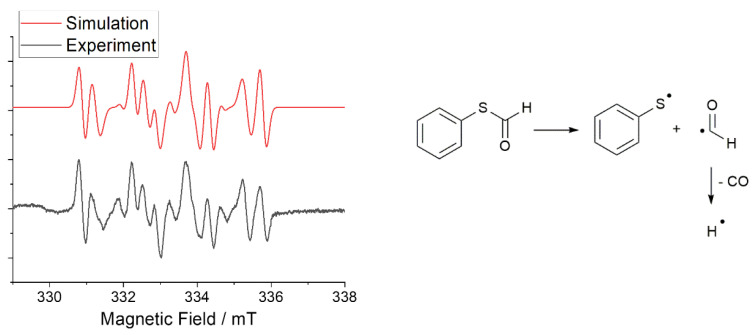
(**Left**): experimental (grey) and simulated(red) EPR spectrum of S-phenyl thioformate photocleavage in the presence of the spin trap DMPO. (**Right**): Mechanism of the generation of thiyl, formyl, and hydrogen radicals by the photocleavage reaction of S-phenyl thioformate.

**Figure 3 polymers-15-01647-f003:**
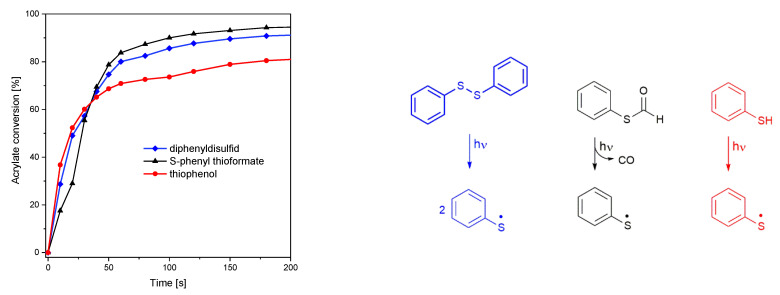
(**Left**): conversion of the hexyl acrylate under UV illumination (185–665 nm, 58.7 mW cm^−2^). (**Right**): photocleavage reaction of diphenyl disulfide, S-phenyl thioformate, and thiophenol.

**Figure 4 polymers-15-01647-f004:**
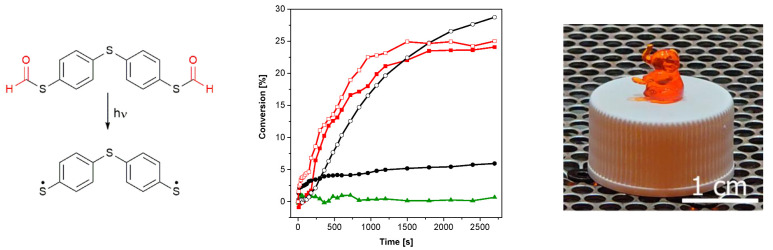
(**Left**): photodecarbonylation reaction of **2b**. (**Middle**): conversion of the acrylate groups of TMPTMA under UV illumination (red line with open square: 1 mM **2b**, 0 days of storage at 50 °C; red line with filled square: 1 mM **2b**, 7 days of storage at 50 °C; black line with open cycle: 1 mM **2a**, 0 days of storage at 50 °C; black line with filled cycle: 1 mM **2a**, 7 days of storage at 50 °C; green line with filled triangles: without PI; 405 nm, 19.15 mW cm^−2^). (**Right**): photograph of the printed test structure.

**Figure 5 polymers-15-01647-f005:**
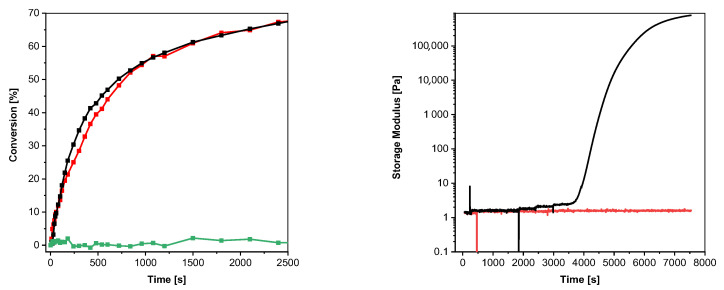
(**Left**): conversion of the acrylate groups of TMPTMA in a TMPTMA/PETMP formulation under UV illumination (red line with squares: 0.14 wt% (1 mM) **2b**, black line with squares: 0.10 wt% (1 mM) **2a**, green line with squares: without PI; 405 nm, 19.15 mW cm^−2^). (**Right**): results of rheological measurements (black line: TMPTMA/PETMP with 0.10 wt% (1 mM) **2a**; red line: TMPTMA/PETMP with 0.14 wt% (1 mM) **2b**).

## Data Availability

Data is contained within the article and Appendix A.

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
