# Peer review of "Exploring Aromatic *S*-Thioformates as Photoinitiators"

_polymers, 2023, doi:10.3390/polym15071647_

Round 1
Reviewer 1 Report
The work is devoted to the study of different thioformates as photoinitiators in the radical polymerization of acrylates under UV and visible light irradiation. The mechanism of photolysis and the main products are explored. The work leaves a good impression, is relevant, and the conclusions are supported by appropriate methods. I believe that the work will be of interest to the readers of Polymers and may be published after some comments and questions have been addressed.
1. page 2, lines 84-86. What method was used to determine the composition of the mixture by HPLC. Simple area normalization? Is this acceptable for a UV detection?
2. Figure 1: The photolysis scheme seems to be not equated for hydrogen. What is the further fate of the H∙ radicals?
3. There are errors with references to figures (page 5, lines 174, 182, 187).
4. Page. 5, line 181: PETMP abbreviation is not defined.
It is a pity that the polymers obtained in the study of photoinitiators efficiency are not characterized in terms of molecular weight distribution. Is it possible to add these results?
Best regards,
Reviewer
Author Response
Dear Reviewer 1,
we are glad that you believe our work should be published. We highly appreciate your comments as we think that they helped to improve the final quality of our manuscript. Please find below our replies to your suggestions:
1. page 2, lines 84-86. What method was used to determine the composition of the mixture by HPLC. Simple area normalization? Is this acceptable for a UV detection?
We have used the areas of the UV-Vis detector signal. Of course, an exact determination of the chemical composition is not possible with this method. We have used this approach for an estimation, which we have now better highlighted in the manuscript.
"The investigation of the photoproducts was carried out by HPLC, determining diphenyl disulfide (appox. 40%) as main product. The area of the UV-Vis detector signals was used to estimate the composition of the product mixture. Besides the educt, i.e. S-phenyl thioformate (approx. 10%), and benzenethiol (approx. 10%) several other photoproducts , which have not been identified, were detected (see Figure S2)."
2. Figure 1: The photolysis scheme seems to be not equated for hydrogen. What is the further fate of the H∙radicals?
You are right - the scheme in Figure 1 on the right does not take into account the H radicals. We have deleted Figure 1 on the right and added the information in the text. The mechanism of the photoreaction is already explained in Figure 2 right.
"The hydrogen radical formed is assumed to undergo recombination reactions with the thiyl radical and the formyl radical, resulting in the formation of benzenethiol and formaldehyde, respectively."
3. There are errors with references to figures (page 5, lines 174, 182, 187).
We have added the correct references in the manuscript.
4. Page. 5, line 181: PETMP abbreviation is not defined.
We defined PETMP in the manuscript.
It is a pity that the polymers obtained in the study of photoinitiators efficiency are not characterized in terms of molecular weight distribution. Is it possible to add these results?
We are planning a follow-up publication based on this communication in which we will cover this topic in detail.
We hope that after the additions and changes made in the manuscript you agree on the publication of this manuscript.
Best Regards,
Thomas Griesser
Reviewer 2 Report
The manuscript by T. Griesser et al. is dealing with the development of a new sub-family of photoinitiators of radical polymerization based on aromatic S-thioformates. This work convincingly demonstrate the potential of S-thioformates providing comprehensive experimental studies using ESR and NMR, both single and kinetic data as well as photopolymerization conversion of acrylate-based 3D printing formulations.
The presentation should be improved: there are several references missing in the text. Also, the full names of organizations in acknowledgement must be provided. Even more important is to provide all experimental details about the experiments. I recommend traditional approach when experimental details and materials/chemicals are listed in a separate chapter and readily available to a reader without the need to refer to SI.
I will recommend the manuscript for publication once the corrections are made.
Author Response
Dear Reviewer 2,
we are glad that you believe our work should be published. We highly appreciate your comments as we think that they helped to improve the final quality of our manuscript. Please find below our replies to your suggestions.
1.) The presentation should be improved: there are several references missing in the text.
You are right - important publications in the field of thiol-based chemistry are missing. Therefore, we have added some further (selected) references on this topic (see references 14-26).
2.) Also, the full names of organizations in acknowledgement must be provided.
We have added the information to the organizations in the acknowledgement.
3.) Even more important is to provide all experimental details about the experiments. I recommend traditional approach when experimental details and materials/chemicals are listed in a separate chapter and readily available to a reader without the need to refer to SI.
Due to the fact that this work is a "Communication" (with limitations in the number of words) we propose to keep it in this form.
We hope that after the additions and changes made in the manuscript you agree on the publication of this manuscript.
Best Regards,
Thomas Griesser